# Application of Enzyme-Linked Fluorescence Assay (ELFA) to Obtain In Vivo Matured Dog Oocytes through the Assessment of Progesterone Level

**DOI:** 10.3390/ani13111885

**Published:** 2023-06-05

**Authors:** Seunghoon Lee, Jin-Gu No, Bong-Hwan Choi, Dong-Kyo Kim, Namwoong Hyung, JongJu Park, Mi-Kyoung Choi, Dong-Hyeon Yeom, Juyoung Ji, Dong-Hoon Kim, Jae Gyu Yoo

**Affiliations:** 1Animal Biotechnology Division, National Institute of Animal Science, Rural Development Administration, 1500, Kongjwipatjwi-ro, Wanju-gun 55365, Jeollabuk-do, Republic of Korea; sage@korea.kr (S.L.); shrkftm@korea.kr (J.-G.N.); narang44@korea.kr (N.H.); pjj7612@korea.kr (J.P.); mimiengel@hanmail.net (M.-K.C.); newgouin@gmail.com (D.-H.Y.); jumoom@gmail.com (J.J.); kimdhhj@korea.kr (D.-H.K.); 2Animal Genetic Resources Research Center, National Institute of Animal Science, Rural Development Administration, 224, Deogyuwolseong-ro, Hamyang-gun 50000, Gyeongsangnam-do, Republic of Korea; bhchoi@korea.kr (B.-H.C.); space1987@korea.kr (D.-K.K.)

**Keywords:** enzyme-linked fluorescence assay, serum progesterone, somatic cell cloning, somatic cell nuclear transfer, dog

## Abstract

**Simple Summary:**

Dog cloning requires in vivo matured recipient oocytes to transfer somatic donor cells for somatic cell cloning. The timing for recovering in vivo matured dog oocytes was determined by predicting the ovulation day based on serum progesterone (P4) concentration in estrus bitches. Radioimmunoassay (RIA) is traditionally used to measure the P4 concentration in dogs. In this study, the P4 concentration for ovulation was measured using next-generation enzyme-linked fluorescence assay (ELFA) and it was compared with that measured using RIA. The oocytes collected in vivo showed a maturation rate of 65.19% after the prediction of ovulation based on the P4 range measured using the ELFA system. These oocytes then produced four cloned puppies. Conclusively, we provide the optimal P4 range for the prediction of ovulation in estrus bitches when ELFA is used to measure the P4 concentration. Thus, we prove the effectiveness of application of ELFA in obtaining in vivo matured oocytes for dog cloning.

**Abstract:**

Successful dog cloning requires a sufficient number of in vivo matured oocytes as recipient oocytes for reconstructing embryos. The accurate prediction of the ovulation day in estrus bitches is critical for collecting mature oocytes. Traditionally, a specific serum progesterone (P4) range in the radioimmunoassay (RIA) system has been used for the prediction of ovulation. In this study, we investigated the use of an enzyme-linked fluorescence assay (ELFA) system for the measurement of P4. Serum samples of estrus bitches were analyzed using both RIA and ELFA, and the measured P4 values of ELFA were sorted into 11 groups based on the standard concentration measured in RIA and compared. In addition, to examine the tendency of changes in the P4 values in each system, the P4 values on ovulation day (from D − 6 to D + 1) in both systems were compared. The ELFA range of 5.0–12.0 ng/mL was derived from the RIA standard range of 4.0–8.0 ng/mL. The rates of acquired matured oocytes in RIA and ELFA were 55.47% and 65.19%, respectively. The ELFA system successfully produced cloned puppies after the transfer of the reconstructed cloned oocytes. Our findings suggest that the ELFA system is suitable for obtaining in vivo matured oocytes for dog cloning.

## 1. Introduction

Since the birth of the first somatic cell cloned dog, “Snuppy”, in 2005 [1], the demand for dog cloning has risen steadily because cloned dogs may replicate elite working dogs or deceased pets [2,3]. The artificial reproduction technology for dogs has been developed to meet this demand. However, significant development in dog cloning technology has not been achieved so far owing to the reproductive characteristics of dogs, which are different from those of other farm animals. A large number of matured oocytes are required to reconstruct a sufficient number of somatic cell nuclear transfer (SCNT) embryos and clone more dogs [4,5]. In vivo dog oocytes are typically used to reconstruct embryos for dog cloning. The collected oocytes require maturation from the moment of collection as in vitro maturation techniques for immature dog oocytes are not yet established [6]. Recent reports on dog oocytes indicate a relatively low rate of in vitro maturation at about 30% compared to approximately 80% in mice and 70% in both sows and cows [7,8,9,10]. One of the reasons for the low in vitro oocyte maturation rate in dogs is a specific physiologic trait in dog reproduction, in which dog oocyte maturation is completed in the oviduct after ovulation, whereas most animals ovulate matured or nearly matured oocytes [11]. In other words, ovulated oocytes are immature and require further stimulation in vitro to completely mature when compared to other mammalian species. For this reason, oviduct flushing and optimal collection times are important for efficient dog cloning.

There are two major methods for determining the optimum oocyte collection time: one based on the range of blood progesterone (P4) levels and the other based on the peak of blood estradiol levels. Immature oocytes typically mature 3 days after ovulation within the oviducts [12]. In the estradiol-peak method, the ovulation day is presumed to be 3 days after the blood estradiol peak; therefore, oocyte collection should be scheduled more than 3 days after the predicted ovulation day [13]. However, this method is rarely used as it requires long-term blood collection over 7 days to detect the estradiol peak. However, any type of analysis system can be used because the decision is based on the peak rather than absolute values. Alternatively, the blood P4-range method is commonly used as it requires a shorter period of daily blood collection (approximately 4–5 days) [14]. However, this method must be altered according to the analysis system as the detected range of blood P4 values differs depending on the analysis system [15,16,17].

The radioimmunoassay (RIA) system for measuring animal blood hormone levels, including P4, was first developed in the 1960s [18]. Although the RIA system has excellent sensitivity, safety problems are associated with the use of radioisotopes as indicators. In the 1970s, the radioisotope indicator was replaced with an enzyme in the enzyme-linked fluorescence assay (ELFA) system [19]. Lastly, an electrochemiluminescence immunoassay (ECLI) system was developed, which has advanced sensitivity and safety [20,21]. These systems are primarily used to analyze blood P4 in animals [22,23,24]. All of the above systems are acceptable in predicting ovulation days of bitches, particularly the estradiol peak method as previously explained. When the P4 level range is used to predict the ovulation day of bitches, the range in each system differs owing to the different sensitivities of each system [25]. In dog species, the P4 level for ovulation is approximately 4–9.9 ng/mL based on the RIA system [26]. Our group investigated the conversion range from RIA to ECLI for P4 levels of bitches. The range of 4–8 ng/mL using a RIA system was adjusted to 6–15 ng/mL using the ECLI system based on measurements of the same samples [15]. However, the broad range of the ECLI system, caused by hypersensitivity, results in lower rates of prediction of ovulation compared to the RIA system, although matured oocytes obtained using the ECLI system have been successfully used to produce cloned dogs.

In this study, the range blood P4 levels of bitches measured using the ELFA system was examined by simultaneously calculating the P4 levels using the RIA and ELFA systems. Lastly, we propose a suitable range of P4 levels for the ELFA system to acquire matured recipient oocytes to be used for successful dog cloning.

## 2. Materials and Methods

### 2.1. Chemicals

Unless otherwise indicated, all reagents were purchased from Sigma Chemical Co. (St. Louis, MO, USA). The measurements using the RIA system was conducted at the Neodin Veterinary Laboratory (Seoul, Republic of Korea) using a DSL-3900 ACTIVE progesterone Coated Tube Radioimmunoassay Kit (Diagnostic systems Laboratories Inc., Webster, TX, USA). All reagents for the ELFA system were purchased from Mini-Vidas automated analyzer (Biomerieux, Marcy-l’Étoile, France).

### 2.2. Animals

For the SCNT study, 2–5 year old mixed-origin large-breed bitches weighing 25–30 kg were used. The bitches were housed in separate temperature- and light-controlled rooms. The bitches were fed a commercial diet twice a day and provided with sufficient water throughout the day. Prior to all surgical procedures, including oocyte collection and cloned embryo transplantation, the bitches were anesthetized using a 9:1 mixture of alfaxalone (alfaxane) and medetomidine (domitor), and then maintained using 2% isoflurane. A pulse oximeter was used to monitor the condition of the bitches [27]. After surgical procedures of embryo collection from oviduct flushing or embryo transfer to oviduct, antibiotics were administered to prevent inflammation. The suture site was checked by veterinarian daily until the wounds were completely healed. Bitches were used as either oocyte donors or surrogate mothers during the experimental period.

### 2.3. Serum P4 Analysis Using RIA and ELFA Systems

Each morning, blood samples were obtained from bitches in the proestrus or estrus phase, and the serum was subsequently divided into two parts for the detection of the P4 level using the RIA and ELFA systems. The RIA system analysis was conducted by the Neodin Veterinary Laboratory. ELFA analysis was performed according to methods described in the user’s manual of the Mini-Vidas automated analyzer. Before analysis, the analyzer was calibrated using calibration reagents provided in the kit.

### 2.4. Oocyte Recovery and Judgement of Oocyte Maturation Status

The cumulus oocyte complexes (COCs) from estrus bitches were recovered by oviducts flushing at 70–72 h after predicted ovulation by serum P4 evaluation. To prevent contamination of COCs, a 1% penicillin/streptomycin mixture was added to HEPES-buffered TCM-199 based flushing medium. The recovered COCs were denuded through pipetting after brief treatment of 0.1% hyaluronidase. Denuded oocytes exhibiting the presence of the first polar body (PB1) were classified as mature. Conversely, oocytes with a perivitelline space exceeding 25 μm and oocytes lacking the PB1 were categorized as ageing and immature, respectively [15].

### 2.5. Donor Cell Culture and Somatic Cell Cloning

Ear skin cells were surgically harvested from male black retriever dogs and promptly transported to the laboratory within 4 h, ensuring a cold environment with ice-cold D-PBS. The tissue underwent three washes with D-PBS, followed by mincing and subsequent culture in advanced Dulbecco’s modified eagle medium supplemented with 10% fetal bovine serum. This cultivation took place at a temperature of 38 °C within a humidified atmosphere consisting of 5% CO_2_ and 95% air. Following 7 days of culture, a fibroblast monolayer was successfully established and was then passaged. For SCNT, cells from passages 3–5 were employed as donor cells. To perform SCNT, denuded matured oocytes were stained with 10 μg/mL of Hoechst 33342 for 5 min. Using an inverted microscope equipped with micromanipulation devices and fluorescence detection filter, the stained PB1 and nucleus of the oocyte were meticulously removed. Subsequently, a donor cell was injected into the perivitelline space of each enucleated oocyte. The fusion of these couplets was accomplished through two pulses of DC electric stimulation at 3 kV/cm for 15 μs. The fused couplets were then activated via calcium ionophore treatment for 5 min, followed by culturing in mSOF medium supplemented with 1.9 mmol/L 6-DMAP for a duration of 4 h [28].

### 2.6. Somatic Cell Cloned Embryo Transfer and Pregnancy Diagnosis

The activated SCNT embryos after electric fusion were transferred into the ampullary portion of the oviducts of surrogate bitches. Surrogate bitches were also selected by P4 evaluation. Estrus bitches which had represented predicted ovulation at 70–72 h before were selected as surrogates in order to synchronize uterus circumstances between oocyte contributors and surrogates. The diagnosis of pregnancy was established by ultrasound detection approximately 31 days post embryo transfer and continued to be monitored until delivery.

### 2.7. Microsatellite Analysis for Certification of Cloned Puppies

A microsatellite is a repeated DNA region in the genome with characteristics of a high mutation rate. This high mutation rate causes genetic diversity compared to other areas of the whole genome. Therefore, microsatellite analysis is often used to certify genetic identification. In this study, microsatellite analysis was employed to confirm whether newborn puppies were cloned individuals. A total of 21 canine microsatellite markers were selected using an annealing temperature of 61 °C, as well as a product size and type of dye that satisfied the condition of multiplex PCR. The multiplex PCR had a 25 μL reaction volume, consisting of 6 μL (20 ng/μL) of genomic DNA, 0.4 μL (10 pmole) each of forward and reverse fluorescence dye primer, 1 μL (unit/μL) of Hot Start Taq DNA polymerase, 4 μL of 10× buffer, and 3 μL of 2.5 mM dNTP. A Thermal Cycler PTC-0240 (MJ Research, Inc., Waltham, MA, USA) multiplex PCR was used according to the following protocol: 15 min at 95 °C for initial denaturation; five cycles of denaturation at 95 °C for 60 s, annealing at 62 °C for 75 s, and elongation at 72 °C for 60 s; five cycles of denaturation at 95 °C for 60 s, annealing at 61 °C for 75 s, and elongation at 72 °C for 60 s; 25 cycles of denaturation at 95 °C for 60 s, annealing at 60 °C for 75 s, and elongation at 72 °C for 60 s; a final extension at 65 °C for 30 min. The PCR products were analyzed using the ABI-3730XL genetic analyzer (Applied Biosystems, Waltham, MA, USA) and GeneMapper version 4.0 (Applied Biosystems).

### 2.8. Statistical Analyses

Statistical analyses were conducted using Prism 5 (GraphPad), and the significance of the P4 levels (ng/mL) was evaluated using a *t*-test. The significance level was set at *p* < 0.05.

## 3. Results

### 3.1. Differences in Blood P4 Level According to RIA and ELFA Systems

To compare the serum P4 concentrations measured by the RIA and ELFA systems, 201 serum samples collected from 69 proestrus or estrus period bitches were analyzed. The average blood P4 levels were analyzed using the two systems simultaneously (Table 1). The serum samples were categorized into 11 groups on the basis of their concentrations determined using the standard RIA system. Each group contained data from 5–84 samples analyzed using RIA and ELFA. Results showed that the two systems exhibited larger differences when the P4 levels were higher, although all groups showed significant differences between the testing systems (*p* < 0.05).

### 3.2. P4 Level in Estrus Bitches Using RIA and ELFA Systems Based on Predicted Ovulation Day

To examine P4 changing tendency and represent the estrus curve of P4 level in estrus bitches, the P4 levels of 45 blood samples taken from 10 random bitches were sorted according to the predicted ovulation day using the RIA and ELFA systems. These samples were sorted by each day from 6 days before the predicted ovulation day (D6) to 1 day after the predicted ovulation day (D + 1) (Figure 1). The average P4 level using RIA was 6.24 ± 0.18 ng/mL, which corresponded to an average value of 9.71 ± 0.69 ng/mL in ELFA on the predicted ovulation day. Figure 1 shows that the P4 level at D0 was significantly higher than at D − 1. This rapid increase in the P4 level on ovulation day was more obvious in the ELFA system than in the RIA system.

### 3.3. Prediction of Ovulation Based on the P4 Level Detected Using RIA and ELFA Systems

We decided the range of 5.0–12.0 ng/mL of ELFA according to the range of 4.0–8.0 ng/mL of RIA referring to Table 1. To estimate the accuracy rate of prediction of ovulation on the P4 ranges determined using the RIA and ELFA systems, we surgically collected oocytes 3 days after predicted ovulation day based on the P4 range measured using both systems and then judged their ovulation status. A total of 61 bitches (43 bitches using RIA, and 18 bitches using ELFA) were selected as oocyte recipients on the basis of the ranges of each system (4.0–8.0 ng/mL and 5.0–12.0 ng/mL in RIA and ELFA, respectively). A total of 546 oocytes (393 and 153 oocytes using RIA and ELFA, respectively) were recovered. The average number of oocytes per bitch was 9.12 and 8.5 for RIA and ELFA, respectively. The rates of acquired matured oocytes were 55.47 and 65.19% for RIA and ELFA, respectively (Figure 2).

### 3.4. Dog Cloning Using In Vivo Matured Oocytes Based on ELFA

To verify whether the ELFA system could be used for dog cloning, we used it with the newly derived P4 range of 5.0–12.0 ng/mL to predict the ovulation of oocyte in the estrus cycle of 38 recipient bitches. A total 419 oocytes were collected, and 287 matured oocytes were used for SCNT. After the SCNT procedure, 165 reconstructed SCNT embryos were finally transferred to 17 recipients (Table 2). Four recipient bitches were confirmed to be pregnant 30 days after transplantation and ultrasound pregnancy tests showed full-term development at 60 days. A total of four cloned puppies were born, but only three survived (Table 2). The genetic identities of the cloned dogs were tested using microsatellite analysis of genomic DNA taken from the somatic cell of the donor dogs, cloned dogs, and surrogates. Analysis of 21 canine-specific microsatellite makers confirmed that the cloned dogs were genetically identical to their respective donor dog as genetic types of 21 microsatellite of all cloned dogs perfectly coincide with donor dog. Both donor dogs and cloned dogs also represented the same phenotypes of hair color, ear shape, etc. (Figure 3).

## 4. Discussion

Successful dog cloning requires a sufficient number of matured oocytes as recipients. In dogs, in vivo oocytes are used as in vitro oocyte maturation techniques are not yet established. A recent report showed that the in vitro maturation rate of dog oocytes is only 30% [7]. One reason for the difficulty in establishing an in vitro maturation technique is that dog oocytes mature in the oviduct after ovulation and take about 72 h from ovulation to complete maturation [29,30]. For the successful collection of mature oocytes by oviduct flushing, an accurate prediction of ovulation time is crucial. The prediction of ovulation day in estrus bitches is typically conducted through the progesterone analysis of serum. However, the measured P4 values may vary depending on the diagnostic system used for analysis. Each analysis system must be calibrated with the appropriate P4 range for that system to accurately predict the ovulation day. In this study, we compared the RIA and ELFA systems to establish the adjusted range of the ELFA system, using the RIA range as a reference for detecting P4 concentration. The P4 levels of the same samples were simultaneously analyzed using the RIA and ELFA systems. We categorized these samples into 11 groups on the basis of their P4 concentration obtained using the RIA system. Although the low concentration groups such as groups 0–1 showed significant differences between the RIA and ELFA systems, at higher concentrations (groups 2–10), the differences observed between the systems increased substantially. ELFA had higher measured concentration levels compared to RIA, which was similar to our previous results that compared RIA and ECLI, which showed larger differences between the two systems at higher P4 levels [15]. Although we did not directly compare ELFA and ECLI, the differences between ELFA and RIA were more moderate than between RIA and ECLI. In another species such as human, some kinds of hormonal levels are detected at a higher range in the ECLI system than in the ELFA system, as well as in the ELFA system than in the RIA system [31,32,33], indicating that the sensitivity of the systems has improved over time. In this study, considering the standard curve of P4 based on range and estrus day, a P4 range of 4.0–8.0 ng/mL based on RIA that corresponded to the ovulation day was measured at 5.0–12.0 ng/mL using ELFA. The range of 5.0–12.0 ng/mL for ELFA was determined according to the range of 4.0–8.0 ng/mL observed in RIA, as shown in Table 1. To account for the empty P4 ranges between groups 3 and 4 in ELFA, we utilized the maximum range of group 3 (4.81 + 0.31 ng/mL) as the minimum range in the ELFA P4 range. On the basis of the same criteria, we employed the maximum range of group 8 (11.10 + 0.56 ng/mL) as the maximum range in the ELFA P4 range. Consequently, although the exact derived P4 range in ELFA was 5.12–11.66 ng/mL, we adjusted the P4 ranges to 5.0–12.0 ng/mL to align with integer for practical application. The 5.0–12.0 ng/mL range of ELFA agrees with the previously reported range of 5.0–10.0 ng/mL [34]. However, their results showed similar levels using both RIA and ELFA, which could be due to different brands of RIA systems being used. The brand of the ELFA system was the same as that used in this study. The applicability of set ranges across brands may be examined in further studies.

After recovering oocytes according to the predicted ovulation day of each system, the adjusted P4 range of ELFA yielded a 65.19% maturation rate of oocytes compared to 55.47% in RIA. This higher maturation rate in ELFA than RIA indicates that ELFA can be used to harvest suitable in vivo matured oocyte for dog cloning. We also showed that matured oocytes collected using the ELFA system for P4 level analysis were successfully used to clone dogs. The increased maturation rate obtained using the ELFA system is because of the steep increase in the P4 level from D − 1 to D0, which is more easily detectable using ELFA owing to the broader range in this system compared to the RIA system. However, a broad range of P4 concentration, as in the ECLI system, can complicate the prediction of ovulation day because the measured value often exceeds the defined range owing to its hypersensitivity at higher concentrations. Indeed, the accuracy of prediction using ECLI is lower than the RIA system [15].

In the estradiol peak method mentioned in Section 1, the hormone range is not factored as only the “up and down” pattern is needed to predict the ovulation day. Therefore, the ECLI system with hypersensitivity is more suitable for the estradiol peak method than the ELFA system. However, the estradiol peak method requires a long examination period, often taking more than 7 days, as the estradiol peak usually forms 72 h before ovulation and requires at least 3 days of data for determination. The P4 range methods are more economical and intuitive. Although P4 analysis for dog cloning has traditionally been conducted using the RIA system, it poses safety concerns because of the use of radioisotopes. Therefore, we examined the applicability of a next-generation diagnostic system, ELFA, for dog cloning. Our findings suggest that a P4 level of 5.0–12.0 ng/mL that was obtained using the ELFA system can be used to determine the ovulation day of dog oocytes. This P4 range in ELFA is broader than that of RIA and narrower than that of ECLI, easily enabling the detection of P4 elevation the day before ovulation and mitigates errors caused by accelerated elevation, particularly at higher concentrations of P4. Because of the potential dangers associated with the use of radioisotopes, there has been a shift from traditional RIA to ELFA or ECLI analysis. Although the newer ECLI system is more suitable for the E2 peak prediction method because of its enhanced sensitivity, the system is not as cost-effective or time-efficient compared to the P4 range method. Therefore, we recommend the ELFA system, which has a reasonable range for the application of the P4-range method, owing to its cost and tome effectiveness. In this study, we confirmed that the ELFA system is safe and accurate and, hence, suitable for obtaining in vivo matured oocytes for dog cloning.

## 5. Conclusions

The precise prediction of ovulation day in estrus bitches is crucial for obtaining a sufficient number of in vivo matured oocytes for dog cloning. A P4 concentration range of 4.0–8.0 ng/mL was determined to indicate ovulation day using the traditional RIA system, whereas it was 5.0–12.0 ng/mL for the next-generation ELFA system. ELFA is a promising and accurate tool for obtaining in vivo matured oocytes for dog cloning.

## Figures and Tables

**Figure 1 animals-13-01885-f001:**
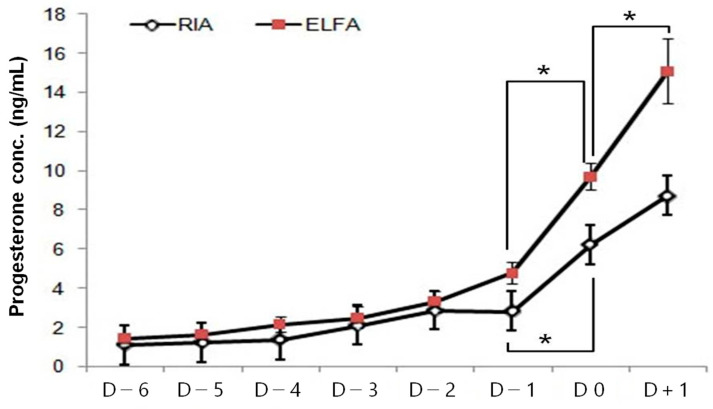
Tendency of average P4 values according to RIA and ELFA systems based on predicted ovulation day. Average P4 levels from D − 6 to D + 1 using RIA (white) and ELFA (red). P4 concentrations of 45 samples derived from 10 bitches were analyzed using RIA and ELFA simultaneously. * The value of P4 is significant at *p* < 0.05.

**Figure 2 animals-13-01885-f002:**
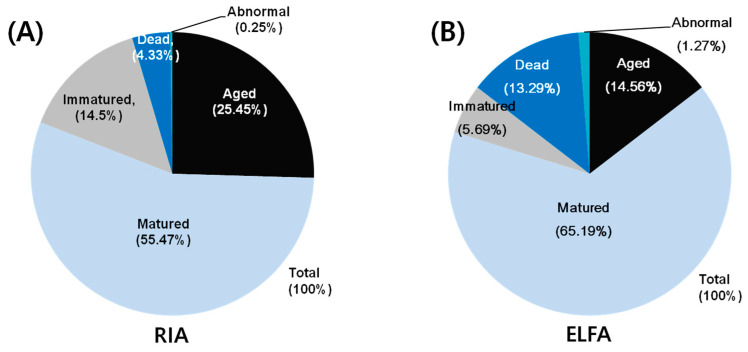
Status of recovered oocyte based on the prediction of ovulation day using the RIA and ELFA systems: (**A**) status of recovered oocytes using a P4 range of 4.0–8.0 ng/mL based on RIA; (**B**) status of recovered oocytes using a P4 range of 5.0–12.0 ng/mL based on ELFA.

**Figure 3 animals-13-01885-f003:**
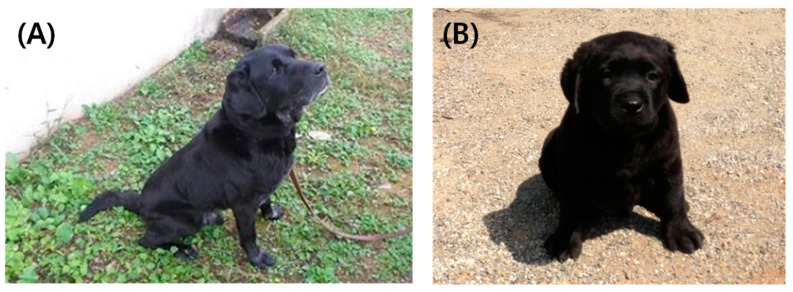
Somatic cell donor black retriever dog and its cloned dog: (**A**) original dog who donated his ear fibroblast cell for cloning; (**B**) one of the cloned puppies at 3 weeks after birth.

**Table 1 animals-13-01885-t001:** Comparison of average P4 values of RIA and ELFA systems in groups sorted according to the average RIA range.

Groups (by P4 Ranges of RIA)	No. of Samples	RIA *	ELFA *	*p*-Value
0	84	0.41 ± 0.03	0.69 ± 0.04	<0.0001
1	24	1.51 ± 0.06	1.57 ± 0.14	0.0304
2	25	2.48 ± 0.06	3.67 ± 0.29	0.0002
3	10	3.43 ± 0.12	4.81 ± 0.31	0.0006
4	10	4.52 ± 0.08	7.19 ± 0.81	0.0041
5	5	5.45 ± 0.18	9.8 ± 1.34	0.0123
6	9	6.48 ± 0.101	9.91 ± 0.53	0.0001
7	12	7.45 ± 0.068	11.16 ± 0.91	0.0005
8	12	8.46 ± 0.05	11.10 ± 0.56	0.0001
9	5	9.66 ± 0.05	16.63 ± 1.67	0.0006
10	5	10.47 ± 0.15	18.83 ± 2.34	0.0074

* Mean ± SEM.

**Table 2 animals-13-01885-t002:** Results of dog cloning performed using in vivo oocytes acquired based on P4 prediction using ELFA.

No. of Transferred Embryos	No. of Recipients	Day 30 Pregnancy (%)	Full Term (%) ^1^	No. of Puppies Born (%) ^2^	No. of Puppies Survived until Weaning (%) ^3^
165	17	4 (23.5)	4 (23.5)	4 (2.4)	3 (75)

^1^ Percentage calculated on the basis of the number of recipients receiving embryos. ^2^ Percentage calculated on the basis of the number of embryos transferred. ^3^ Percentage calculated on the basis of the number of puppies born.

## Data Availability

Not applicable.

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
