# Peer review of "Application of Enzyme-Linked Fluorescence Assay (ELFA) to Obtain In Vivo Matured Dog Oocytes through the Assessment of Progesterone Level"

_animals, 2023, doi:10.3390/ani13111885_

Round 1
Reviewer 1 Report
this manuscript is informative and applicable for dog cloning. I wish the authors to emphasize or describe the advantage of ELFA compared to the traditional ways and the differences between them in the discussion section.
the authors need to clarify without repeating. for example
"in vivo matured recipient oocytes are required for dog cloning" would be better" in simple summary
recommended abstract as follow "Successful dog cloning requires a sufficient number of in vivo matured oocytes as recipient oocytes for reconstructing embryos. Accurate prediction of ovulation day in estrus bitches is critical for collecting mature oocytes. Traditionally, a specific serum progesterone (P4) range in the radioimmunoassay (RIA) system has been used for ovulation prediction. In this study, we investigated the use of an enzyme-linked fluorescence assay (ELFA) diagnostic system for P4 measurement. Serum samples from estrus bitches were analyzed using both RIA and ELFA, and the measured P4 values were compared on the day of ovulation. The ELFA and RIA prediction ranged from 5.0–12.0 ng/ml and 4.0–8.0 ng/ml of P4, respectively. The rates of acquired matured oocytes in RIA and ELFA were 55.47% and 65.19%, respectively. Both RIA and ELFA methods successfully produced cloned puppies after the transfer of reconstructed cloned oocytes. Our findings suggest that the ELFA diagnostic system is suitable for obtaining in vivo matured oocytes for dog cloning."
Supposedly, in Figure 1, closed and open are better than colored squares and diamond shapes.
line 222 because rather than as
the authors need to clarify without repeating.
please referer to my comments above
Reviewer 2 Report
There are some questions the author need to explain in the article:
1、How to deal with the dogs after surgically obtained the oviducts?
2、The uniformity and repeatability of mixed- breed dog is not so stable,why not experiment Beagle dog?
3、The 206 dogs used in this experiment were all 5-year-old, 25-30kg?
4、What’s the cloning effiency acquired using RIA prediction? The ELFA P4 prediction efficiency was low.
5、Result of 3.3, for ELFA prediction, only 153 eggs were recovered, but in result of 3.4, 165 eggs were transferred by ELFA prediction?
6、How did the author deal with the oocytes obtained from RIA prediction(393)?
Reviewer 3 Report
This paper aims to compare two methods (radioimmunoassay and an enzyme-linked fluorescence assay) to diagnose progesterone levels in bitches in order to predict ovulation and further improve mature oocyte collection from the oviduct. I consider that the title could be revised, since it could be misleading not to refer P4 determination on it.
Line 68: Not enough evidence, since you refer only one paper which compared two methods
Line 101: To avoid misleading information, it is better to use bitches instead of dogs from this point on
Line 177: Rewrite, it seems that you only collected two samples, one on day-6 and other on day +1
Line 179: could significance be marked in figure 1 somehow? Differences between both measurement systems are significant?
Line 181: Is this significant? What is the p value?
Line 195: In my opinion, this is one of the aspects to improve in this paper. It is not explained how this new interval is reached, is it through the values of table 1? What is the methodology?
Line 213: Further explanation needed. The four bitches were pregnant with only one foetus each? All three surviving puppies were confirmed to be cloned dogs?
Again, in discussion it is advised to explain deeply how the values 5.0-12.0ng/mL were reached
Although writing is generally OK, some editing of English language is advised in sentences such as “one of cloned dogs” in line 218.
Round 2
Reviewer 2 Report
No.
Author Response
Response to Academic Editor’s Comments
Dear Academic Editor.
Thank you for decision of minor revision. We carefully reviewed and fixed about entire manuscript according to your comments.
Point 1: The English language still require editing..
Response 1: Before the first submission, we already had English correction by Editage company.
We would like to attach the correction certificate, however the system is not allowed. Even that, we conducted english editting one more time based on your comment.
Point 2: Please specify further the methodology of nuclear transfer and cloning, if the methos used for the progesterone evaluation is relevant for cloning specifically or would be the same in relation to any other ARTs
Response 2: Especially in dog cloning, ovulation prediction by progesterone evaluation is required to acquire in vivo matured oocytes as a recipient oocytes for nuclear transfer and cloning. However, other domestic animals generally used in vitro oocytes for cloning because in vitro oocyte maturation protocol has been established. We added the methodology for oocyte recovery by progesterone evaluation for dog cloning (line 139-140).
Point 3: regarding oocyte maturation, use up-to-date references
Response 3: We all updated references about oocyte maturation. Reference number 7-10 was changed to up-to-date references. According to recent result of dog oocyte in vitro maturation in reference 7, in vitro dog oocytes maturation rate fixed from 7.8% to 30% in manuscript in introduction and discussion (line 62, line 296).
Point 4: change ml to mL and ul to uL throughout
Response 4: We changed ml to mL and μl to μL in entire manuscript.
Point 5: set the number of bitches used in each experiment
Response 5: The number of bitches were confirmed and added to line 227 for experiment of Table 1 (Comparison of average P4 values of RIA and ELFA system in groups sorted by average RIA range). When calculating number of bitches, we found that number of serum sample was wrong in manuscript. Serum sample number was corrected ‘206 samples’ to ‘201 samples’ in manuscript (line 227) and ’10 samples’ to ‘5 samples’ in group 9 of Table 1 (line 246). Fortunately, changed sample number did not affect to the average and standard deviation of P4 values in group 9 of Table 1. Because, the values of 5 samples had been just doubled before correction. About the number of bitches in each experiment of Figure 1, Figure 2, and Table 2, we have already written down in manuscript.
Point 6: the protocol of anesthesia requires references
Response 6: We added reference 27 regading with the protocol of anesthesia (line 120, 455) and changed the name of anesthesia from brand name to generic name (line 118).
Point 7: specify further the methodology of nuclear transfer and cloning
Response 7: We added some details during nuclear transfer and cloning process (line 140-144, 168-172, 187-191) and added the reference 28 of dog somatic cell cloning (line 177, 457).
Point 8: L141, needs references
Response 8: We added references (Reference 15) about classifying dog oocyte stage (line 147).
Point 9: L166-L179. Please explain the microsatellite analysis and its results
Response 9: We added explaination of the microsatellite analysis for certification of cloned dog (line 200-204) and supplemented its results (line 281-284).
Point 10: Please do not repeat what was said in the Introduction in the Discussion section
Response 10: We deleted repeat in discussion section (line 301-304, 307-308, 310-312).
Thank you for your effort and cooperation.
Sincerely yours,
Seunghoon Lee.
